# Predicting Cyberbullying Perpetration in US Elementary School Children

**DOI:** 10.3390/ijerph20156442

**Published:** 2023-07-25

**Authors:** Christopher P. Barlett

**Affiliations:** Department of Psychological Sciences, Kansas State University, Manhattan, KS 66506, USA; cpb6666@ksu.edu

**Keywords:** cyberbullying, children, cellular phone, online harm

## Abstract

Cyberbullying has emerged as a societal issue, and the majority of the research examining cyberbullying perpetration samples adolescent and/or emerging adult populations. A paucity of empirical attention has focused on young children (aged 8–10) regarding their cyberbullying frequency and predictors. The current study sampled 142 US youth aged 8–10 years and assessed their cyberbullying perpetration rate and cellular phone ownership. Results indicated that (a) older participants were more likely to cyberbully than their younger peers; (b) higher rates of cyberbullying were found for youth who already owned a cellular phone; and (c) an interaction between participant age and cellular phone ownership was found, suggesting that cyberbullying was highest for only the 10-year-old group who owned a cellular phone. These findings have implications for (a) parents, school administrators, health care providers, and anyone else interested in better understanding the predictors of cyberbullying perpetration; (b) intervention specialists focused on reducing cyberbullying in youth; and (c) a researcher interested in understanding the basic theoretical underpinnings of cyberbullying. Based on these findings, we recommend that (a) cyberbullying interventions be administered to youth as early as elementary school; (b) parents/guardians carefully consider the positive and negative consequences of youth cellular phone usage; and (c) increased communication between youth and parents/guardians concerning youth cellular phone activities.

## 1. Predicting Cyberbullying Perpetration in US Elementary School Children

Cyberbullying is defined as repeated intentional harm delivered to another where a power imbalance is present using electronic means [1,2]. Results from a systematic review of the literature show that 10–60% of youth are cyber-victimized and 6–32% are cyberbullies [3]. The variability in these percentages is likely due to artifacts and moderators between the studies synthesized, including the age of the population, measure to assess cyberbullying perpetration and victimization, the year the data were collected, and others [4,5]. Moreover, meta-analytic findings show the deleterious psychological effects of being cyber-victimized and engaging in cyberbullying perpetration, such as anxiety, depression, low self-esteem, and loneliness [6]. Such findings highlight the need to better understand the variables that predict cyberbullying perpetration. 

The majority of research examining the predictors of cyberbullying perpetration samples adolescent or emerging adult populations without considering young children (≤10 years old). We focused our analysis on two important variables: cellular phone ownership and participant age. We are unaware of any published research focused on the main effects and interaction between cellular phone ownership and participant age in a sample of US youth aged 8–10 years old. Empirically studying the variables that predict cyberbullying perpetration is important for (a) gaining knowledge about the scope of the “cyberbullying problem”; (b) understanding what variables increase the likelihood of harming others online; (c) developing a theory focused on the processes germane to cyberbullying; and (d) creating or adapting intervention programs to reduce cyberbullying perpetration. The past two decades have seen a welcomed and expanding literature addressing these issues to yield many great theoretical and applied strides. However, important gaps in the literature remain. Indeed, the ever-changing technological landscape necessitates continued research that delineates a better understanding of these issues—especially in a sample of young child populations. To fill this gap, we sampled US children (aged 8–10 years) and asked them to complete questionnaires used to assess their cyberbullying perpetration experiences and cellular phone ownership.

## 2. Cyberbullying in Young Children

Much of the accumulated knowledge regarding cyberbullying perpetration has come from studies that sampled adolescent or emerging adult populations. Targeting these populations is appropriate considering the research showing that cyberbullying perpetration frequency increases from childhood (ages 11–17) to emerging adulthood (ages 18–26 [7])—a weak, albeit significant, meta-analytic finding (*r* = 0.05, 95% CI: 0.03, 0.08 [6]). However, substantially less is known about cyberbullying in populations of youth below 10 years of age. Indeed, Bauman and Bellmore [8] reviewed the literature and noted that, “…we envision research that expands the study of cyberbullying across the lifespan. Such research could help us understand how young children and older adults (both arguably vulnerable groups) who use technology are managing their digital relationships and contexts” (p. 6). Agreeing with this quote, our focus was to examine the predictors of cyberbullying in a sample of youth between 8 and 10 years of age. 

The few published studies sampling youth between 8 and 10 years have shown that cyberbullying perpetration is infrequent. For instance, data collected in young children showed an average cyberbullying score of 0.89 (on a 0–9 scale) indicating that youth do not often engage in cyberbullying at this age [9]. Englander [10] sampled 8–10-year-old children and found that 5.8% cyberbullied their peers. In addition to cyberbullying perpetration being infrequent, research has also showed no sex differences in cyberbullying perpetration within this age range [9,11]. Overall, past research has suggested that elementary school-aged children do not cyberbully often; however, the percentage (or average) cyberbullying score suggests that there are some instances of cyberbullying occurring in populations of young children. While uncommon, there are still youth between 8 and 10 years of age who harm others online. Moreover, the majority of the published research sampling youth in elementary school is dated, and there are likely temporal changes that may influence the number of youths who cyberbully others. While our study cannot address this issue specifically, we contend that studying and understanding cyberbullying in this age group is critical for at least two reasons:

First, from a developmental perspective, young children (on average) are beginning to form and automatize aggressive scripts, aggressive schemas, and aggressive attitudes that form after continued experiences with and exposure to violence [12]. Cyberbullying is one type of aggression [13], and research has shown a large effect size between cyberbullying perpetration and aggression-related constructs (e.g., normative aggressive beliefs: *r* = 0.37, and anger: *r* = 0.20 [6]). Therefore, youth between the ages of 8 and 10 may be one of the most important age groups to study due to the malleability of their aggressive-cognitive development. In addition, cyberbullying theory—one that includes developmental and learning-based tenets—contends that initial experiences as a cyber-aggressor eventually lead to the development and automatization of consistent and repeated cyberbullying behavior through additional positively reinforced cyber-aggressive actions that develop anonymity perceptions, cyberbullying attitudes, and cyberbullying beliefs [13]. Since young children are unlikely to have a history of cyberbullying other individuals [10], understanding cyberbullying at young ages is imperative. 

Second, survey data show that young children have access to Internet-capable devices. Data collected from over 2000 UK children aged 5–16 show that 53% of children who own a cellular phone did so by the age of 7 years [14]. However, no published study that we are aware of has tested the interaction between participant age and cellular phone use on cyberbullying perpetration. Englander [10] sampled over 4500 8–11-year-old US youth and found that cellular phone ownership predicted cyberbullying perpetration—cyberbullying was more likely for those who owned a cellular phone. However, Englander (2018) did not report on the possible interaction between age and cellular phone ownership on cyberbullying perpetration, which begets the purpose of the current study. Data from the Pew Research Center suggest that the amount of engagement with smartphones in youth aged 0–11 (assessed via parent report) increased from 59% for 5–8-year-old youth to 67% for 9–11-year-old youth [15]. Overall, younger youth are being allowed to have their own cellular phone, which likely allows for increased Internet access to these individuals. Moreover, focused on a narrower sample of youth aged 8–10, cellular phone use increase with age, and the current study will examine whether age differences and cellular phone ownership interact to predict cyberbullying.

## 3. Overview of the Current Study

The purpose of this current study was to examine the role that participant age and cellular phone ownership predicted cyberbullying perpetration in a sample of young US children (aged 8–10 years). Based on past research and theory, we predict that cyberbullying perpetration will be the highest for older (compared to younger) youth who do (compared to those who do not) own a cellular phone. 

## 4. Method

### 4.1. Participants

We recruited 142 (57.2% male) 3rd and 4th grade school students to participate in the current study from three local elementary schools in the same school district in Southern Pennsylvania, US. The average age of the sample was 8.89 (SD = 0.73) years (age range 8–10 years; N_8_ = 43; N_9_ = 60; N_10_ = 28; 11 did not report their age). The majority of participants were in fourth grade (57.4%). An a priori power analysis in G*Power showed that 134 participants were needed to achieve 0.95 power with an alpha = 0.05, two-tailed test, with an effect size (r) of 0.30—obtained from Englander [10]. 

### 4.2. Procedure

Permission to conduct the study was granted by the author’s former Institutional Review Board. The superintendent of the area school district, the school board, and building principals granted permissions. In Fall 2019, an informational letter was sent to parents of 3rd and 4th grade children explaining that the study was interested in assessing children’s technology use, online attitudes, and cyberbullying behaviors. We obtained active parental consent electronically or via paper–pencil forms. In early December 2019, the primary researcher arrived at each elementary school and greeted the participating children in a large common room. The researcher explained to the participants that (a) the study was about their online behaviors and attitudes; (b) all responses were anonymous and confidential; and (c) participation was voluntary. We then obtained child assent to participate. Because of the age of the youth, the primary researcher provided two practice questions to facilitate competence with the various rating scales (one with a three-point scale and the other with a five-point scale). After the practice questions were completed, the participants completed the following questionnaires before being thanked. 

## 5. Materials

### 5.1. Cyberbullying Perpetration

We assessed cyberbullying perpetration using a researcher-created questionnaire adapted from Englander [10] and Doane and colleagues [16]. This 11-item questionnaire asked participants to indicate how often in the past year they engaged in various online behaviors using a 1 (*never*) to 5 (*everyday or almost everyday*) rating scale. We defined the term “online” for participants as, “Online is defined as when you are on the Internet. You can access the Internet from a computer, cell phone, video game system, tablet, and other devices” to clearly differentiate these behaviors from in-person behavior. Due to the exploratory nature of this questionnaire on a sample this young, we conducted an exploratory factor analysis (EFA) to determine the factor structure of the items. Early analysis showed that one item, “I have made a mean picture about another online”, needed removed because it loaded onto its own factor. Results (with this item removed) showed that the ten remaining items loaded onto two factors accounted for 71.90% of the variance (see Table 1). The item, “I sent a rude message to another online”, was removed because it cross-loaded onto both factors. The remaining nine items loaded onto two factors: specific cyberbullying behaviors (five items; α = 0.88; sample item: “I used a bad word at another person online”) and general cyberbullying behaviors (four items; α = 0.80; sample item: “I have been mean to someone online”). We summed the items that corresponded to each factor such that higher scores indicate higher levels of cyberbullying perpetration. 

### 5.2. Demographics

A demographic questionnaire assessed sex, age, and grade in school. We did not assess other demographics (e.g., ethnicity, SES, living situation) due to ethical reasons. In addition, we asked participants whether they owned their own cellular phone, measured by a 1 = yes; and 0 = no response scale. 

There are many other variables that can influence cyberbullying behavior besides cellular phone use and participant age [6]. We attempted to study other predictors of cyberbullying. Indeed, additional questionnaires were administered to participants that assessed cyberbullying attitudes [17], anonymity perceptions [18], cyberbullying beliefs [19], and technology access [20]. However, these measures have never been used on a sample this young, and, therefore, low internal consistency with these measures was found. Therefore, the responses to these unreliable questionnaires were not used.

### 5.3. Data Analysis Plan

First, we will conduct two sets of preliminary analyses that will inform our primary analyses. This will include (a) correlating both types of cyberbullying in order to obtain an understanding of the overlap between general and specific cyberbullying perpetration and (b) conducting independent sample *t*-tests to examine sex differences in cyberbullying to determine whether we can account for them in our primary analyses. In order to examine the prevalence rates of cyberbullying in a young US sample, we will examine the distributional properties and central tendency of both types of cyberbullying. Then, we will test the age of the participant (3 levels: 8-, 9-, and 10-year-old) by cellular phone ownership (2 levels: yes, no) interaction on specific and general cyberbullying perpetrations. 

## 6. Results

### 6.1. Preliminary Analyses

First, both types of cyberbullying correlated highly (*r* = 0.64, *p* < 0.001), as expected. Second, independent samples *t*-tests were conducted to examine the sex differences in specific and general cyberbullying. The results showed no difference between the male and female participants in the specific cyberbullying perpetration, *t*(123) = 1.12, *p* = 0.26, or general cyberbullying perpetration, *t*(126) = 0.72, *p* = 0.47. 

### 6.2. Specific Cyberbullying

The average score on the cyberbullying specific scale was 5.56 (SD = 2.08) with a possible range of 5–25. Moreover, 82.8% of our sample scored the lowest possible score, indicating no involvement in cyberbullying perpetration. Figure 1 displays the histogram to show the skewed (*Z* = 32.98, *p* < 0.001) and kurtotic (*Z* = 144.73, *p* < 0.001) distribution of cyberbullying scores.

A 3 (age: 8; 9; 10-year-old) × 2 (cellular phone ownership: yes; no) analysis of variance (ANOVA) was conducted with specific cyberbullying as the outcome. Results show a significant main effect of age, *F*(2, 113) = 4.58, *p* = 0.012, *η^2^_p_* = 0.075. Pairwise comparisons with a Bonferroni correction showed that 10-year-old participants (M = 6.74, SE = 0.41) cyberbullied significantly (*p* = 0.004) more than 8-year-old participants (M = 5.15, SE = 0.35), but no differences emerged when comparing the 9-year-old group (M = 5.62, SE = 0.30) to the other two age groups. Results also show a main effect of the cellular phone ownership, *F*(1, 113) = 7.49, *p* = 0.007, *η^2^_p_* = 0.062, such that those who owned a cellular phone (M = 6.39, SE = 0.33) cyberbullied more than those who did not own a cellular phone (M = 5.28, SE = 0.24). However, these results were qualified by a significant two-way interaction, *F*(2, 113) = 5.37, *p* = 0.006, *η^2^_p_* = 0.087. Simple effects analysis showed that the main effect of cellular phone ownership was not significant for the 8-year-old youth, *F*(1, 113) = 0.18, *p* = 0.67, or 9-year-old youth, *F*(1, 113) = 0.81, *p* = 0.37; however, there was a significant main effect of cellular phone ownership for the 10-year-old youth, *F*(1, 113) = 14.70, *p* < 0.001, *η^2^_p_* = 0.115, such that cyberbullying perpetration was higher for the 10-year-old youth who owned a cellular phone (M = 8.30, SE = 0.64) compared to those who did not own a cellular phone (M = 5.19, SE = 0.50). Figure 2 displays this interaction.

### 6.3. General Cyberbullying

The average score on the general cyberbullying perpetration scale was 4.52 (SD = 1.33) that had a possible range of 4–20. Moreover, 80% of the sample reported the lowest score on this scale, indicating that cyberbullying was infrequent. The similarity in the findings in general and specific cyberbullying perpetrations is expected because these two measures were significantly correlated, *r* = 0.64, *p* < 0.001. Figure 3 displays the histogram to highlight the skewed (*Z* = 14.95, *p* < 0.001) and kurtotic (*Z* = 26.29, *p* < 0.001) distribution of the cyberbullying scores.

The same 3 (age) × 2 (cellular phone ownership) ANOVA with general cyberbullying perpetration showed a significant main effect of age, *F*(2, 116) = 5.36, *p* = 0.006, *η*^2^*_p_* = 0.085. Pairwise comparisons with a Bonferroni correction showed a significant difference (*p* = 0.001) between the 8-year-old (M = 4.16, SE = 0.22) and 10-year-old groups (M = 5.27, SE = 0.26), but no differences were found when comparing the 9-year-old group (M = 4.58, SE = 0.19) to the other two age groups. Also, a significant main effect of cellular phone ownership, *F*(1, 116) = 3.99, *p* = 0.048, *η^2^_p_* = 0.033, emerged, such that those who owned a cellular phone (M = 4.93, SE = 0.21) cyberbullied more than those without a cellular phone (M = 4.41, SE = 0.15). However, the interaction was not significant, *F*(2, 116) = 2.81, *p* = 0.064. 

## 7. Discussion

The purpose of the current study was to examine the prevalence of cyberbullying perpetration in a sample of young (aged 8–10 years) US children and test whether age and cellular phone ownership predicted cyberbullying perpetration. Our results suggest that cyberbullying perpetration—whether specific or general—is relatively infrequent at this age. The inspection of the mean score on either cyberbullying scale juxtaposed with the percentage of youth who reported never cyberbullying suggests that cyberbullying is uncommon—a finding consistent with past work [10]. However, we caution readers in assuming that cyberbullying is not important at this age group based on these results. First, while the majority of youth in our sample do not cyberbully, a non-trivial proportion had engaged in some cyberbullying in the past year. The inspection of our percentages shows that approximately 17% (specific cyberbullying) and 20% (general cyberbullying) engaged in cyberbullying at least once in the past year. These rates are a substantial increase from the research published years ago on US youth aged 8–11 (5.8% [10]). We are aware that such comparisons may be ill-advised considering the different measurement tools used to assess cyberbullying. Regardless, our findings suggest that cyberbullying perpetration is occurring in populations of young US children.

Results from our study also suggest that participant age and cellular phone ownership influence cyberbullying (specific and general), and these two variables interact to influence specific cyberbullying perpetration. Older youth in our sample (10-year-old participants) were more likely to cyberbully than younger youth (8-year-old participants). Moreover, cyberbullying was more likely if a youth owned a cellular phone, which replicates Englander [10]. Interestingly, we found an interaction between participant age and cellular phone ownership on specific cyberbullying perpetration—the highest cyberbullying perpetration rate occurred in only the 10-year-old sample that owned a cellular phone. From a developmental perspective, we are unsure why differences between the 8-year-old and 10-year-old sample emerged, and future research should test the mediating or moderating variables that explain these differences. For instance, research has shown lower prosocial behavior [21] and slightly higher indirect aggression [22] in a 10-year-old sample compared to an 8-year-old sample. Moreover, Auxier et al. [15] found higher cellular phone use with older, compared to younger, samples of US youth. However, other mediators and moderators are likely.

### 7.1. Theoretical and Applied Extensions

Results from the current study offer important theoretical insights. The Barlett Gentile Cyberbullying Model (BGCM [19]) posits that the first time an individual uses online communication to harm another is a learning trial, in which that individual likely perceives themselves as anonymous and believes that their physical stature (e.g., muscles, height) are moot. Continued positively reinforced cyber-aggressive behaviors automatizes these perceptions and beliefs to lead to the development of positive cyberbullying attitudes that are the key predictor to subsequent cyberbullying perpetration [13]. In the current study, two key findings emerged that have important BGCM applications. First, 8–10-year-old youth do not cyberbully often, which suggests that studying the developmental tenets of BGCM could be accomplished in youth population. Indeed, Barlett [13] noted that the developmental postulates of BGCM are understudied, and important theoretical questions remain, including (a) the number of cyber-aggressive trials needed to automatize and develop anonymity perceptions and cyberbullying beliefs, (b) how early reinforcement and/or punishment for initial cyber-aggressive behaviors changes the cyberbullying learning, and (c) the moderating role of personality or environmental contexts that influence cyberbullying learning. One way to answer these research questions is to longitudinally follow youth who have never engaged in cyberbullying and monitor their cyber-aggressive actions over time while also examining BGCM variables. Results from this study suggest that younger children (especially 8–10 years old) are an important population to study.

Second, the results show that the age of the child interacted with the cellular phone ownership to influence cyberbullying perpetration. Cellular phone ownership only predicted cyberbullying perpetration in the 10-year-old sample. If we assume that owning a cellular phone provides youth with a means by which to engage in cyber-aggressive behaviors, then the BGCM can account for these effects. Research has shown that technology use and access predict BGCM tenets in a sample of emerging adults [20]. Although we did not assess the mediating processes that explain why cellular phone use predicts cyberbullying perpetration in the current study, we can presume that cellular phone ownership increases one’s self-efficacy to cause online harm. Interestingly, this effect was only found in 10-year-old youth, and not their younger peers. Although speculative, perhaps changes in (a) parental monitoring of media [23]; (b) knowledge about Internet complexity [24]; (c) importance and function of peer relationships [25]; and (d) myriad cognitive developmental processes (e.g., verbal working memory and visual spatial working memory [26]), play an important role. Juxtaposed with owning technology that affords the user the ability to harm another online, this creates an environment suitable for cyberbullying. BGCM has the theoretical flexibility to account for these findings, and future research should test these important mediator and moderator variables.

Finally, the findings also have implications for applied intervention efforts aimed at reducing cyberbullying. Myriad meta-analyses have shown that cyberbullying intervention curricula are successful at reducing cyberbullying perpetration [27,28,29,30]; however, we are unaware of any intervention that targets youth aged 8–10 years. Results from the current study suggest that approximately 20% have engaged in cyberbullying, which necessitates interventions targeting this population. We do not advocate for trading any traditional bullying instruction with cyberbullying lessons; however, we do suggest that intervention specialists adapt and utilize cyberbullying reduction curricula in elementary schools. Hopefully, future research can be successful at creating such interventions. Moreover, interventions that target parent education regarding cyberbullying have been shown to be successful [31]; however, our results suggest that such education needs to address cellular phone ownership in the lessons.

### 7.2. Limitations and Future Research

Limitations exist in the current research that necessitate future research. First, we assessed other cyberbullying-related constructs, such as anonymity perceptions and cyberbullying attitudes; however, poor reliability negated the use of these assessments. The lower than optimal reliability is likely a function of trying to adapt scales validated on older populations to younger children. We are unaware of any anonymity perceptions or cyberbullying attitude measure normed and validated on a youth sample. Future research should attempt to create and validate cyberbullying-related assessments on a youth sample and attempt to test cyberbullying theory. Second, we cannot make causal claims due to the correlational nature of the data. Our study was originally approved to be a two-wave longitudinal study; however, access issues due to school safety protocols related to COVID-19 required us to stop the study before Wave 2 data collection (which was scheduled to occur December 2020). Nevertheless, future research should continue to examine the predictors of cyberbullying in youth with longitudinal designs to establish causality. Third, we did not assess variables that may moderate or mediate our observed relationships. Future research should measure parental monitoring, general media use, technology access, and other relevant variables to further examine why and for whom our observed effects are found. Finally, our research findings are only generalizable to the population sampled in the current study—mostly white US youth. Future work should attempt to replicate our findings using a more diverse population of youth, which would likely necessitate a higher sample size than the one used in the current study (*n* = 142). 

### 7.3. Closing Remarks

Overall, cyberbullying perpetration is occurring in a non-trivial percentage of youth aged 8–10 years. Juxtaposed with the decreasing age of cellular phone ownership [14], it is important for parents, healthcare providers, school administrators and teachers, and youth to understand the behavioral ramifications of early technology access. We believe that our research is an important first step in understanding the initial causes of cyberbullying that future research should build upon. 

## Figures and Tables

**Figure 1 ijerph-20-06442-f001:**
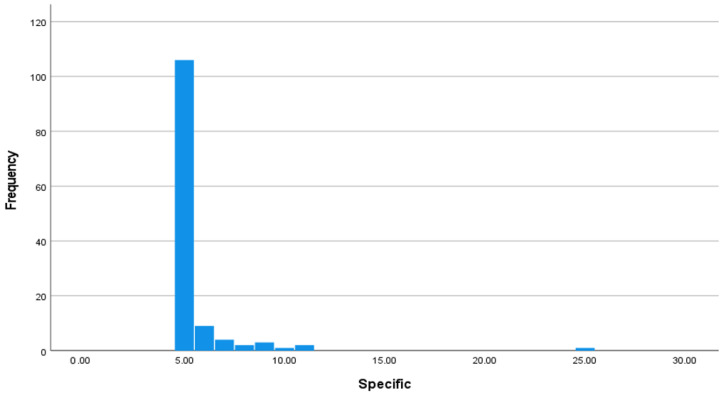
Histogram of cyberbullying specific scores.

**Figure 2 ijerph-20-06442-f002:**
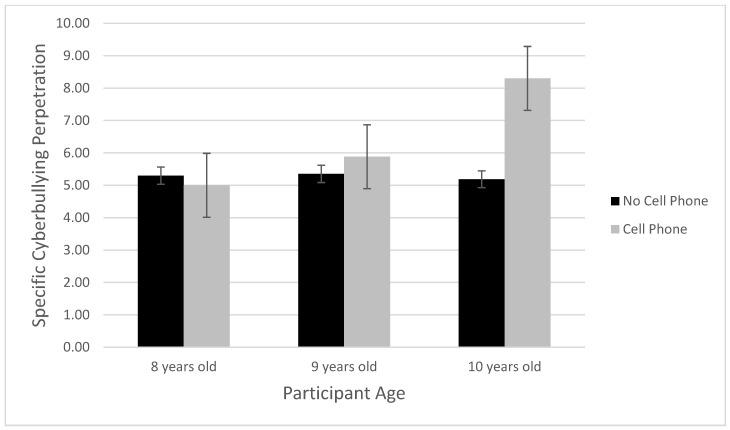
Age × cell phone ownership interaction.

**Figure 3 ijerph-20-06442-f003:**
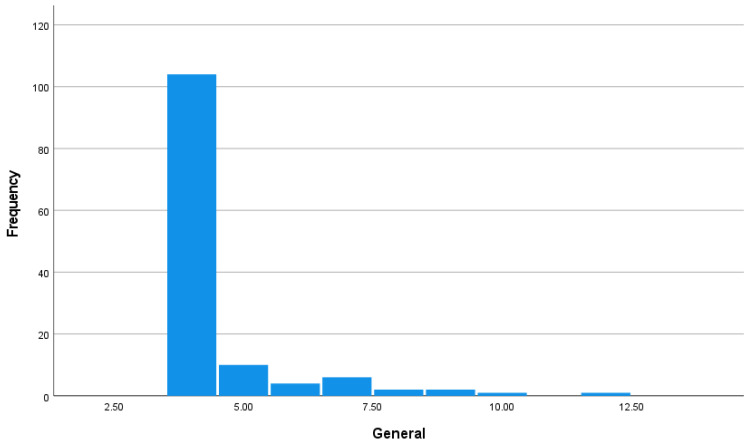
Histogram of cyberbullying general scores.

**Table 1 ijerph-20-06442-t001:** Factor loadings for each item in cyberbullying perpetration measure.

Item	General	Specific
I have made up a lie about another person and told others online	0.822	
I have sent mean messages or a mean question to another online	0.832	
I called someone mean names online	0.647	
I have posted something mean about another on Instagram	0.962	
Twitter, Facebook, or other social media program		
I have sent other people mean messages online		0.613
I used a bad word at another person online		0.633
I made fun of another person online		0.791
I teased someone online		0.808
I have been mean to someone online		0.814

## Data Availability

Data is available upon request.

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
