# Peer review of "Predicting Cyberbullying Perpetration in US Elementary School Children"

_ijerph, 2023, doi:10.3390/ijerph20156442_

Round 1

Reviewer 1 Report

this study examined the cyberbullying perpetration rate and its relation with age and  cellular phone ownership, which seems to be interesting However some majore concerns were not well addressed:

(1) the resaerch question is not clear. for example, why did you focused on youth aged 8-10 years? especially thet are not active in online activities;

(2) 142 participants were a small simple,, whcih is a great limitation;

(3) except for age and cellular phone ownership, thre are other important factors influencing  cyberbullying  of children, why didi you were concerned with age and cellular phone ownership? and this issue was not clearly discussed in the introduction;

(4) the ecessity to conduct this study is not clear;

(5) the analysis is also too simple, and the examinnation on the interaction of age and cellular phone ownership was also not pointed out clearly;

(6) the discusion is also weak

Reviewer 2 Report

Dear Editor

Dear author

Thank you for inviting me to review this paper. Interesting paper about cyberbullying. This research may provide a significant contribution in the field of psychology at this time. Below you can find my comments to improve the manuscript:

Abstract section: please add the contributions of this study, the implications, and recommendations according to your research findings.

In this interesting topic, I am from the position of the reader, wanting a detailed discussion of bullying and cyberbullying and an in-depth explanation of existing research.

We need introduction and literature review separated. a more specific and detailed explanation is needed.

Methodology section: author need to add data analysis section.

References: I found inappropriate self-citations (5 citations) in this article. it is good to be careful of this behavior. I suggest author to change the citations and references, consider there is many studies about cyberbullying.

Reference must be improved. How many references can proof how deep author understand and master this topic.

Round 2

Reviewer 1 Report

I appreciate it very much for your careful revision and response. It's a contribution to focus on  Elementary School Children. However, there are obvious limitations for the research design (as your  response stated  that "we also measured anonymity perceptions, cyberbullying attitudes, and cyberbullying beliefs; however, the internal consistency was too low for those questionnaires to be included in this paper", this may indicate design defect).

Author Response

Thank you for this comment. We believe that the issue is not a design issue. Indeed, the design is a correlational design, which is commonplace in this field. However, the bigger issue here is the lack of measurement for youth of this age. We noted, in the limitations, that better measures need validated on a sample this young. Therefore, we feel that the design is appropriate and acceptable; however, the poor reliability of the measures used was the bigger issue. 

Reviewer 2 Report

Author has revised the manuscript well. The manuscript ready to published.

Well Done.

Author Response

Thank you for your thorough review.